# Legislation on Medical Assistance in Dying (MAID): Preliminary Consideration on the First Regional Law in Italy

**DOI:** 10.3390/healthcare13091091

**Published:** 2025-05-07

**Authors:** Lorenzo Blandi, Russell Tolentino, Giuseppe Basile, Livio Pietro Tronconi, Carlo Signorelli, Vittorio Bolcato

**Affiliations:** 1School of Public Health, Vita-Salute San Raffaele University, 20132 Milan, Italy; 2Centre for Evidence-Based Medicine, University of Oxford, Oxford OX1 2JD, UK; 3National Centre for Epidemiology and Population Health, Australian National University, Canberra 2601, Australia; 4Department of Biomedical Sciences and Public Health, University “Politecnica delle Marche” of Ancona, 60121 Ancona, Italy; 5Department of Health and Life Sciences, European University of Rome, 00163 Rome, Italy; 6Scientific Directorate, Maria Cecilia Hospital, GVM Care and Research, 48033 Cotignola, Italy; 7Maria Beatrice Hospital, GVM Care & Research, 50121 Firenze, Italy

**Keywords:** medical assistance in dying, end of life, assisted suicide, assisted dying, ethics, law

## Abstract

Medical assistance in dying (MAID) remains a sensitive and evolving issue in Europe, frequently linked with discussions about human freedom, life dignity, and healthcare policy. While national consensus in Italy is absent, the Region of Tuscany has enacted Law No. 16/2025, which establishes a MAID procedure based on recent Constitutional Court rulings. The commentary aims to provide a preliminary analysis of the new law, addressing ethical, medico-legal, and social issues that emerge in relation to the Italian and global debate on the topic. The law establishes a three-stage process based on four eligibility criteria: irreversible disease, psycho-physical suffering, life-support dependence, and informed consent. However, Tuscany’s model poses medico-legal and ethical concerns, particularly about the boundaries of regional legislative competence, the duties of healthcare professionals, and the possibility of intra-national inequity or “health migration.” In addition, critical organisational implications derived from informed consent and lethal drug self-administration impede clinical implementation in some individuals with mental or neurological disorders. The lack of clarity in the different steps of the procedure, the uncertain supervision system, and the potential consequences for specific categories of vulnerable people underline the need for comprehensive national regulation. A future regulatory framework must balance procedural clarity with individual autonomy and equitable access, bringing Italy in line with larger European context for end-of-life care.

## 1. Background and Legal Context

The issue of medical assistance in dying (MAID) or assisted suicide [1] has become one of the most debated topics in public health legislation across Europe, along with vaccination and digitisation policies, all of which are closely related to the right of freedom and the dignity of life [2]. Recent comparative legal reviews by Orsini et al. and Ciliberti et al. have highlighted the lack of uniformity in the requirements, options, timing, and procedures to access MAID worldwide [2,3]. The dignity of life, including end of life, and the right to make autonomous decisions in health are extensively debated across societies, where cultural, religious, clinical, and philosophical perspectives influence how life and death are perceived. The “slippery slope” of access to MAID, especially in complex medical conditions, calls for appropriate and clear policies on information and regulation, to safeguard the right to freedom in its most comprehensive sense [4].

Several jurisdictions have established clear and ethically sound approaches to MAID that can inform Tuscany’s regional framework and contribute to a consistent national policy in Italy, balancing individual autonomy, clinical responsibility, and public accountability to reduce the risks of misuse and ethical violations [5,6]. For example, Canada’s revised MAID law under Bill C-7 introduced a two-track system: Track 1 for cases where death is foreseeable, allowing expedited access with fewer procedures, and Track 2 for non-foreseeable cases, requiring enhanced safeguards like 90-day assessments and specialist consultations [7]. Belgium permits euthanasia under stringent legal conditions that require evidence of constant and unbearable suffering, with cases reported to the Federal Control and Evaluation Commission for Euthanasia to ensure compliance and prevent abuse. The Netherlands has established regional committees to review every case to ensure clinical practices align with legal standards and include protocols for sensitive cases like dementia or psychiatric disorders [8].

In Italy, given the absence of specific regulating law, the Italian Constitutional Court had intervened on this matter soliciting Parliament. By two key rulings—22 November 2019, no. 242 and 18 July 2024, no. 135—the Constitutional Court justified the fact of “aiding someone in suicide” under some conditions, which are viewed as the necessary requirements to initiate the MAID process, thus decriminalising it [9]. However, despite these rulings, Parliament have continued to grapple with ethical, medico-legal, organisational, and, perhaps most significantly, cultural aspects [2,6].

The current practice on end-of-life context in Italy was widely and recently discussed according to Constitutional Court opinions, arguing through some high-profile cases life-sustaining treatment refusal or withdrawal (Law 219/2017, art. 1), palliative care and pain therapy (Law n. 38/2010 and Law 219/2017, art. 2), unbearable suffering or ineffective treatment definition, and the healthcare provider’s role [10,11,12,13,14].

In this context, the Italian Region of Tuscany enacted the Law 14 March 2025 no. 16 “Organisational modalities for the implementation of the Constitutional Court rulings 242/2019 and 135/2024” (translated with the support of a professional software set for juridical text and reported in the Appendix A), which outlined the procedure for granting citizens access to MAID.

This article proposes a preliminary analysis of the new legislation, examining the ethical, medico-legal, and social implications that arise within the framework of both the Italian and global context.

In a civil law country, the effort made by the regional legislator to initiate a discussion based on a law, its application, and any possible amendments must certainly be highlighted. The law grounds on the Italian Regions’ legislative competence over the organisational aspects of healthcare activities. With the direct reference to the four requirements outlined by the Constitutional Court, Tuscany developed a procedure for MAID that is both unique and significant in the national debate. Particularly, Parliament could draw on the regional one to establish a national regulation, or consider further initiatives, while the Constitutional Court could be asked to conduct a legitimacy assessment of the law.

The four necessary conditions are mentioned in the preamble of the law. Those individual’s conditions are as follows: (i) the presence of an irreversible disease, (ii) the presence of unbearable psycho-physical suffering, (iii) the individual’s dependence on life-support treatment, and (iv) the individual’s ability to make free and informed decisions.

The law proposes a three-step procedure lasting 37 days with the option for a one-time extension of 5 days from the initial application by the individual [15].

The first phase lasts 20 days, starting with the individual’s application for MAID to the competent local health authority. A multidisciplinary commission (hereinafter Commission) (article 3) ascertains the requirements on provided clinical records. Within this period, the Commission may seek a further opinion from the Committee of Clinical Ethics (hereinafter Committee). Furthermore, article 5 on the verification of the requirements states that the Commission verifies applicant adequate information about the possibility of accessing a palliative care pathway, the right to refuse or withdraw consent to any health treatment, including life support, and the possibility of accessing deep palliative sedation. Then, a response is provided to the individual who can define the protocol for MAID, such as the drugs to be used and the location for MAID implementation. Then, in the second phase, the protocol is verified by the same Commission and by the Committee within 10 days, and an answer is given to the applicant. After the validation, the third phase involves the implementation of the protocol over a 7-day period, culminating in the self-administration of the lethal drug.

## 2. Discussion

### 2.1. Cultural and Ethical Issues Due to Territorial Differences

While the Italian Parliament has not yet reached a shared solution [16], a debate has emerged about whether MAID is really an effective legislative competence of the Region, after legislative devolution, or it clearly goes beyond a mere issue of organising regional healthcare services [16]. However, what truly matters is that, given the specific reference to the competent local health authority, it remains unclear whether all Italian citizens are entitled to access the practice in Tuscany [17]. Both the scenarios could lead to critical consequences: firstly, a MAID-related migration or unequal rights for citizens could occur across Italy; secondly, the access-to-service problems could arise from ethical and cultural barriers. The unequal accessibility of healthcare across Italian regions—in terms of both methods and waiting times—particularly in the fields of oncology and rare diseases, could lead to inequitable access to this practice or even to its misuse [18,19].

Indeed, a similar long-debated discussion took place for the voluntary pregnancy interruption by the Law 22 May 1978, no. 194 “Rules for the social protection of maternity and voluntary pregnancy interruption” [20]. This issue of conscientious objectors was so widespread that many gynaecologist-obstetricians refused to perform the procedure, prompting the definition of reference centres with non-objecting professionals to guarantee access to the service, as a broader public health issue [21].

### 2.2. Guidelines and Standardisation of Protocols

The pathway of the Tuscany Law on MAID is also quite challenging to implement. A very important issue is that it would be more reasonable for the local health authority to develop a set of standardised protocols, giving individuals the option to choose after being properly informed. This would help ensure uniformity and allow the validation of protocols through the Italian National Bioethics Committee and/or health scientific societies, such as the palliative care’s one, as far as for palliative sedation [11,22]. A clearer approach to guideline modelling could improve and secure the process. In addition, some components of the Commission could be the same as the facilities Committee, creating complexity and redundance. Additionally, it is not detailed whether the Committee must always be involved or only in difficult cases, as a second opinion. It is important to note that the Constitutional Court 242/2019 ruling mentions the competent Ethical Committee—regulated by the national law—for further requirement verification, and not the Committee of Clinical Ethics—which is established at the regional level within healthcare facilities [23].

It is worthy of in-depth discussion whether the ruling 242/2019 effectively grants the right for MAID with the assistance of the facilities of the Italian National Health Service (NHS). In fact, it merely exempts actions of aiding someone in suicide from punishment. The ruling places on the NHS the responsibility to determine the conditions for non-punishability, the appropriate execution method, and the choice of drugs. The Tuscany Law 16/2025, however, explicitly refers to healthcare professionals’ assistance. The adult palliative sedation guidelines, published in 2023 and currently being updated in a new 2024 version by the Italian National Institute of Health, do not provide for the participation of healthcare professionals and NHS facilities in MAID procedure [17]. The guidelines distinguish between the medical assistance involved in the procedure of deep palliative sedation, aimed at relieving suffering by inducing a state of deep unconsciousness until natural death, and, on the other hand, the mere preparation of drugs and devices for MAID, in the absence of ongoing healthcare and monitoring.

### 2.3. Alternative Options to MAID and Deep Sedative Palliation (DPS)

Tuscany’s regional Law 16/2025 highlights the need to disclose to the patient before verifying the requirements the alternative options of MAID, such as refusal or withdraw medical treatments, including broadly life support, artificial nutrition and hydration, or the possibility of accessing palliative care until continuous deep palliative sedation (DPS) [24]. Moreover, the mentioned deep palliative sedation is not truly an alternative of MAID [25]. The requirements for undergoing deep palliative sedation are the existence of an unfavourable short-term prognosis or imminent death, where the doctor must refrain from any unreasonable persistence in administering care and avoid unnecessary or disproportionate treatments [26]. However, apart from patient specific refusal, artificial hydration and nutrition—by law included in medical treatments—can be continued and may be in place [27]. The third requirement involves a suffering that is refractory to medical treatments, in which case the doctor may resort to continuous deep palliative sedation in conjunction with pain management therapy. The fourth condition is that the patient’s consent must be obtained.

For MAID, the pathological condition must be merely “irreversible”, without any further prognostic indication; as for neurodegenerative disorders, it may be long-term and slowly progressive. What is particularly relevant, then, is the lack of overlap between the two procedures, from the clinical conditions to the legal framework. However, to exclude this option, the clinical condition must be analysed by the Commission, although this preliminary evaluation remains somewhat undefined within the procedure described. Likewise, if the discussion of possible alternatives to MAID or other care path were omitted or inadequately disclosed to patients, it remains unclear by whom and how it would be carried out.

Difficulties in clearly defining ineffective treatments or their lack of proportionality, refractory pain or unsustainable suffering are current challenges for procedures, both clinically and legally, which then impact the verification of the requirements, as it depends on the interests involved and on those who hold or represent them [28]. Indeed, minimal prospects for improvement or minor qualitative changes may be interpreted differently by the patient, their family, or the healthcare professional. Many questions also remain regarding the dependence on life-sustaining treatments; indeed, the law and Court rulings parallel the refusal to initiate such treatments, when necessary, with their withdrawal, even if in clinical practice they are quite different [12,29].

The key difference between the two procedures, legally speaking, lies in the behaviour of healthcare professionals. In DPS, physicians (a medical act) actively manage the drug and the care to alleviate unbearable and refractory pain until death, and it is strictly linked with the short-term prognosis of an irreversible disease that must be certified by a physician (Law 219/2017, art. 2) [30]. On the other hand, in MAID, healthcare professionals (mainly nurses) only limit themselves in predisposing the drugs and devices for the self-administration by the patient [26]. This distinction is crucial, as otherwise active behaviour in lethal drug administration by healthcare providers in Italy contrasts with the Constitutional Court rulings. In fact, only “passive” support for suicide is decriminalised, while the “active” one remains forbidden and punishable.

Another issue is the option of the arrangement of living wills for end-of-life procedures, particularly in disorders with progressive evolution and affecting decision-making ability, in Italy known as shared care planning (SCP). In the MAID case, the patient must be of sound mind, not only for informed consent disclosure but also because they are required to self-administer the lethal drug. Then, the option of SCP becomes impractical, though legally valid, in the context of palliation. In fact, the patient would not be able to self-administer the drug when the clinical condition outlined in the shared care planning is realised, as they could be incapable. On the contrary, in cases of deep palliative sedation, this option is viable as the physician could start the procedure when the patient is no longer able to express consent or is in a state of incapacity, according to the registered shared care planning between the patient and the physician [31].

Thus, the need for mental capacity and physical ability means that all patients with a mental or neurological degenerative condition must proceed with MAID before they become incapable, determining a somehow acceleration in the decision-making process, while it requires careful consideration [32].

### 2.4. Waiting Time

The waiting times, as seen in cases of voluntary pregnancy interruption, should be considered, especially as they are linked to information provision by the health and social services. Similarly, in this case, the Commission or other experts in the field should ensure information and alternative option disclosure. Otherwise, how is it possible or practical to assess the patient’s understanding? In the specific case, it could replace and/or integrate the protocol validation period. The waiting or “reflecting” time must be a separate period to the timing of the procedure as in the British draft, contrarily to the one-month procedure in the Netherlands [33].

However, this period still represents a time of extreme suffering for the patient, and therefore, it is necessary to balance the need for a period of reflection and the proper unfolding of the procedure with the imperative to minimise suffering.

This period varies significantly across the world, depending on differing legal and medical requirements [6]. In Italy, where there is no short-term prognosis or imminent risk of death seeking assisted dying, a one-week reflection period—with medical information and possible family involvement—could be beneficial.

### 2.5. Psychiatric Conditions and MAID

MAID for psychiatric conditions remains particularly critical, as the “subjective” requirement is prevalent, and it is necessary to measure and assess consistently [34,35]. According to current Italian regulation on advanced directives, shared care planning and deep palliative sedation, integrated by the current Tuscany law, it is extremely difficult to apply the procedure in relation to psychiatric conditions, particularly with regard to the dependence on life-sustaining treatment criterion. Hence, the access to MAID for psychiatric patients seems not viable [36]. The Belgian and Dutch regulations in fact, in defining the eligibility requirements for the procedure, offer a broader framework than the Italian model currently being discussed: a clinical condition that is irremediable or progressive or incurable, unbearable suffering, and the absence of therapeutic alternatives [33]. In such cases, the issue of assessing mental capacity in the decision-making process would also arise, as well as the individual’s ability to ultimately carry out self-administration, unlike in Benelux countries where euthanasia is legal, and a protocol exists for verifying compliance with legal standard [8].

## 3. Conclusions

In Italy, Tuscany is both the first and only region to enact a law on MAID. The implementation of this law undoubtedly represents a milestone, although it has raised numerous organisational, medico-legal and ethical issues. It is debated whether this topic falls under regional legislative competencies or State prerogatives. Ambiguity may lead to territorial inequalities in the practice. Tuscan law does not provide standard protocols of MAID, and the Commission’s role in ascertaining the requirements to access is unclear. Ethical concerns arise regarding the conditions of access required, and the healthcare professionals’ roles within the procedure. Incorporating mechanisms from jurisdictions like Canada, Belgium, and the Netherlands into Italian law could protect vulnerable individuals, support clinical clarity, and foster public trust in ethically sound end-of-life decision-making. In the hoped-for perspective of national legislative attention in MAID, procedural rigour will need to be better balanced with uniform management, respect for individual freedom, and end-of-life dignity within the broader European context.

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
