# Peer review of "Legislation on Medical Assistance in Dying (MAID): Preliminary Consideration on the First Regional Law in Italy"

_healthcare, 2025, doi:10.3390/healthcare13091091_

Round 1
Reviewer 1 Report
Comments and Suggestions for Authors
Thank you for the opportunity to review this opinion piece. The manuscript addresses an ethically and legally sensitive topic of high societal relevance: the regulation and implementation of Medical Assistance in Dying (MAID), especially concerning recent developments in the region of Tuscany. It is commendable that the authors contribute to this important and often controversial debate. Thoughtful and well-balanced solutions in this field can benefit not only patients and families, but also medical professionals and society as a whole.
However, I believe that the manuscript would benefit from improvements in both language and structure to enhance its clarity, accessibility, and argumentative strength. I encourage the authors to accept the following points as friendly suggestions and support to achieve their aim to be heard in the discussion.
Content-related Suggestions
- At the outset, it would be helpful if the authors acknowledged the complexity of the topic more explicitly and recognized the sincere efforts made by regional policymakers to find workable solutions. Such an introductory note could lend greater weight to the subsequent critical analysis.
- The identity and motivation of the authors remain somewhat unclear. Clarifying whether the perspective presented is based on legal, medical, or interdisciplinary experience would help readers better understand the framework of the argument.
- At the end, the article presents a broad spectrum of national approaches, but in the background section, these are outlined in a way that may seem too coarse or generalized. It may help to identify key contrasts or extreme positions, while acknowledging the different priorities (e.g., safeguarding autonomy vs. protecting vulnerable groups) that shape national laws. The section on international legal frameworks (Netherlands, Canada, Belgium, Great Britain), currently at the end of the discussion, could be more effective if integrated into the introductory part of the manuscript, helping to frame the issue from the outset within a broader, comparative context. The discussion could then refer to this context, thus strengthening the critical arguments.
- The comparison to palliative sedation is valuable, yet the text occasionally lacks clarity about which medical procedure is being referenced at a given point (MAID or sedation). This could be resolved through more explicit language or clearer transitions between sections.
Structural Suggestions
- Sections 1 through 3 are currently very short and somewhat fragmented. I recommend merging them under a new heading such as "Background and Legal Context". This would allow for a more fluid narrative and help readers grasp the legal foundations before moving into critical discussion.
- The discussion would greatly benefit from a stronger argumentative structure. I suggest that the main arguments of the opinion piece be clearly stated early in the text and then used as subheadings in the discussion. This would guide the reader and lend coherence to the overall presentation.
This opinion piece raises important questions. I encourage the authors to revise the manuscript and I look forward to reading a more polished and focused version.
Comments on the Quality of English LanguageWhile the manuscript is generally understandable, the English language quality needs improvement. Many sentences appear influenced by Italian syntax, which affects readability. A thorough language revision by a native or near-native speaker is strongly recommended to improve grammar, clarity, and fluency.
Author Response
On behalf of all authors, I wish to resubmit our original article n. healthcare 3604279 titled “Legislation on Medical Assistance In Dying (MAID): preliminary consideration on the first regional law in Italy” after revisions.
We would like to thank the Editors and Reviewers for the comments and suggestions which gave us the opportunity to make the manuscript more consistent and clearer.
We have answered to the reviewers in the specific online section by rewriting Reviewers’ comments in bold, followed by authors answers.
In the manuscript, changes are tracked with Word track changes. To allow better revision readability, the major revisions of single paragraph and those added were shown in red. We report here all reviewers' comments and authors answers.
Thank you for the opportunity to review this opinion piece. The manuscript addresses an ethically and legally sensitive topic of high societal relevance: the regulation and implementation of Medical Assistance in Dying (MAID), especially concerning recent developments in the region of Tuscany. It is commendable that the authors contribute to this important and often controversial debate. Thoughtful and well-balanced solutions in this field can benefit not only patients and families, but also medical professionals and society as a whole.
However, I believe that the manuscript would benefit from improvements in both language and structure to enhance its clarity, accessibility, and argumentative strength. I encourage the authors to accept the following points as friendly suggestions and support to achieve their aim to be heard in the discussion.
Many thanks. We have extensively revised the text and English edited.
Content-related Suggestions
- At the outset, it would be helpful if the authors acknowledged the complexity of the topic more explicitly and recognized the sincere efforts made by regional policymakers to find workable solutions. Such an introductory note could lend greater weight to the subsequent critical analysis.
Many thanks. We have revised the introduction and discussion mentioning regional legislator efforts.
- The identity and motivation of the authors remain somewhat unclear. Clarifying whether the perspective presented is based on legal, medical, or interdisciplinary experience would help readers better understand the framework of the argument.
Many thanks. We have mentioned the perspective of analysis in the introduction.
- At the end, the article presents a broad spectrum of national approaches, but in the background section, these are outlined in a way that may seem too coarse or generalized. It may help to identify key contrasts or extreme positions, while acknowledging the different priorities (e.g., safeguarding autonomy vs. protecting vulnerable groups) that shape national laws. The section on international legal frameworks (Netherlands, Canada, Belgium, Great Britain), currently at the end of the discussion, could be more effective if integrated into the introductory part of the manuscript, helping to frame the issue from the outset within a broader, comparative context. The discussion could then refer to this context, thus strengthening the critical arguments.
Many thanks. We have revised accordingly and developed the introduction also according to the other reviewer’s suggestions.
- The comparison to palliative sedation is valuable, yet the text occasionally lacks clarity about which medical procedure is being referenced at a given point (MAID or sedation). This could be resolved through more explicit language or clearer transitions between sections.
Many thanks. We have revised the paragraph. We have then discussed from a medico-legal point of view the difference between DPS (active medical act) and MAID (passive nursing act) to highlight the limited perimeter of practical implementation of the law.
Structural Suggestions
- Sections 1 through 3 are currently very short and somewhat fragmented. I recommend merging them under a new heading such as "Background and Legal Context". This would allow for a more fluid narrative and help readers grasp the legal foundations before moving into critical discussion.
- The discussion would greatly benefit from a stronger argumentative structure. I suggest that the main arguments of the opinion piece be clearly stated early in the text and then used as subheadings in the discussion. This would guide the reader and lend coherence to the overall presentation.
Many thanks. We have revised accordingly and add sub heading in the discussion.
This opinion piece raises important questions. I encourage the authors to revise the manuscript and I look forward to reading a more polished and focused version.
Many thanks. We have extensively revised the text.
Comments on the Quality of English Language
While the manuscript is generally understandable, the English language quality needs improvement. Many sentences appear influenced by Italian syntax, which affects readability. A thorough language revision by a native or near-native speaker is strongly recommended to improve grammar, clarity, and fluency.
Many thanks. We have extensively revised the text and English edited.
sincerely, livio tronconi
Reviewer 2 Report
Comments and Suggestions for Authors
Section 2: I think a little more context would be useful here. What are the current laws and practices regarding withdrawal of life-sustaining medical interventions? Is this part of the current Italian debate, or does it focus only on providing the patient with a lethal dose of a drug (or similar means of dying)? Have there been high-profile cases of people charged with assisting in suicide? Were they convicted or acquitted?
Lines 52-55: When you say that the court “justified” a crime, do you mean that the court decided that someone was justified in violating the law? Or did the court convict someone of a crime but give a very light punishment because of extenuating or justifying circumstances? Or do you mean that the court provided additional justification for regarding assisted suicide as a crime? Please clarify.
Line 78: “Alleged” is the wrong word here. Perhaps you meant to say that the commission will use the medical records that are provided or the records that are available.
Line 91: I think the question mark here should be a period.
Your discussion falls into 3 different categories, and it would be helpful to keep these separate (perhaps putting them in different sections of the paper): (1) Legal issues regarding the legal authority or “competence” of a regional legislative body to make this rule; (2) Administrative and medical concerns about how this rule could be put into practice; and (3) Ethical arguments about whether MAID (as described in this rule) is ethically permissible. At present, these 3 kinds of considerations are mingled together, and this reduces the clarity and effectiveness of your arguments.
Lines 129-131: Although it’s not entirely clear here, you seem to be arguing that it would be very, very difficult for any patient to meet the requirement of experiencing suffering that is refractory to medical treatment. That is because one of the treatments available is deep palliative sedation – this would suspend all of a patient’s experiences, including any form of suffering. A standard view of continuous deep palliative sedation is that it should be used only when pain is intolerable and refractory to less extreme interventions and death is expected within a short time. (See: de Graeff A, Dean M (2007) Palliative sedation therapy in the last weeks of life: a literature review and recommendations for standards. J Palliative Med 10(1): 67-85.) Would a feeding tube be used in such cases?
Lines 132-134: Please clarify whether the difficulties you describe here are for the deep palliative sedation criteria, the MAID criteria, or both.
There is a great deal of complexity in these issues that is being skimmed over. In assessing whether a treatment is ineffective, for instance, we need to clarify what that treatment is intended to accomplish, how that goal of treatment is related to the patient’s overall goals for treatment, what tolerance the patient (and physician) have for bad outcomes, which low-probability outcomes are worth pursuing, and so forth. The concept of proportionality is also complex and is often found in bioethics scholarship rooted in Roman Catholic theology. If these issues are important to your argument, present some explanation of each.
Lines 136-139: It’s not clear what you mean here. Your description of this law in section 3 did not include a requirement that all medical alternatives to MAID had been ruled out. Please clarify what the law actually requires and revise section 3 to reflect that.
- What is the commission’s role here? Does it make medical determinations about which treatments are viable alternatives to MAID? Does it assess whether enough alternatives have been tried and assess whether they have truly failed?
- By the way, how would an alternative be ruled out? Must the patient actually try every treatment for which s/he is a medical candidate? What if a patient does not consent to one of these alternatives? Does that mean that they cannot meet the requirements for MAID? What if an alternative treatment is too expensive for the patient, or is not available within a reasonable distance? What about the option of voluntary cessation of eating and drinking? (See: Quill TE, Ganzini L, Truog RD, et al. (2018) Voluntarily Stopping Eating and Drinking Among Patients With Serious Advanced Illness—Clinical, Ethical, and Legal Aspects. JAMA Intern Med. 178(1):123-127.)
Lines 140-142: I thought deep palliative sedation was being presented as a difficulty in meeting the requirement that the patient be in intolerable, intractable pain (with the result that no one would ever meet the criteria for MAID). But here you’re addressing the suggestion that deep palliative sedation is an alternative to MAID – in other words, that it offers another means of dying in order to avoid further suffering. Has anyone taken that position? If so, it would be helpful to cite it. If not, why bother to argue against it?
Lines 143-163: This paragraph is very unclear. It begins with a comparison of drugs used in MAID and continuous deep palliative sedation. How is this related to a discussion of problems with MAID? The paragraph then briefly addresses legal differences between active and passive euthanasia, and then turns to the complex issue of informed consent, decision-making competence, and advance directives. Please separate these issues, show how they are relevant to the topic of this paper, and address them in sufficient detail.
Lines 157-163: The issue of informed consent is very important and very complex; it requires a much more detailed discussion.
- What are the legal standards for assessing decision-making capacity and what are the clinical tests used to make this determination?
- Would someone experiencing intolerable suffering still be competent to make this sort of life-and-death decision? Why or why not?
- Are there concerns in MAID cases about whether the patient’s decision is unduly influenced by factors such as the cost of treatment, the availability of treatment and supportive care, the attitudes of family members, concern about “being a burden” to family, etc.?
- What does the law say about whether a patient can use a living will or other advance medical directive to choose MAID?
Lines 164-171: The waiting periods commonly found in MAID laws generally serve 2 purposes: (a) providing time for medical evaluation and consultation, competence assessment, exploring options, etc.; and (b) allowing an opportunity for the patient to reflect on this choice and perhaps decide against it. It is worth pointing out that we are dealing here with patients whose suffering is so great that they would rather die than continue to endure it. The longer we make them wait, the more avoidable suffering they experience. One of the ethical challenges, then, is to find the right balance between providing enough time for reflection of the sort associated with purpose (b) and minimizing suffering by keeping that time as short as possible. In the American state of Oregon, data shows that a number of people who start the process of seeking medical assistance in dying die from their illnesses before that process is complete. (See: https://www.oregon.gov/oha/ph/providerpartnerresources/evaluationresearch/deathwithdignityact/Documents/year26.pdf)
Other ethical issues associated with MAID have not been addressed: The implied devaluing of the lives of people with disabilities or chronic illnesses, the vulnerability for abuse or misuse, etc.
The conclusion (section 5) mentions some issues that have received very little attention in this paper: The issue of access and medical migration, lessons to be learned from laws in other countries, and neurodegenerative or psychiatric diseases. Please expand on these topics in the main body of the paper.
Comments on the Quality of English LanguageThe writing is sometimes awkward and needs clarification in some areas (as noted in my comments).
Author Response
On behalf of all authors, I wish to resubmit our original article n. healthcare 3604279 titled “Legislation on Medical Assistance In Dying (MAID): preliminary consideration on the first regional law in Italy” after revisions.
We would like to thank the Editors and Reviewers for the comments and suggestions which gave us the opportunity to make the manuscript more consistent and clearer.
We have answered to the reviewers in the specific online section by rewriting Reviewers’ comments in bold, followed by authors answers.
In the manuscript, changes are tracked with Word track changes. To allow better revision readability, the major revisions of single paragraph and those added were shown in red. We report here all reviewers' comments and authors answers.
Comments and Suggestions for Authors
Section 2: I think a little more context would be useful here. What are the current laws and practices regarding withdrawal of life-sustaining medical interventions? Is this part of the current Italian debate, or does it focus only on providing the patient with a lethal dose of a drug (or similar means of dying)? Have there been high-profile cases of people charged with assisting in suicide? Were they convicted or acquitted?
Many thanks, we have added a paragraph also quoting relevant literature on the topic.
Lines 52-55: When you say that the court “justified” a crime, do you mean that the court decided that someone was justified in violating the law? Or did the court convict someone of a crime but give a very light punishment because of extenuating or justifying circumstances? Or do you mean that the court provided additional justification for regarding assisted suicide as a crime? Please clarify.
Many thanks, the Court decriminalised the fact if performed under some conditions, so the agent could not be punishable. We have revised the sentence.
Line 78: “Alleged” is the wrong word here. Perhaps you meant to say that the commission will use the medical records that are provided or the records that are available.
Many thanks. We have revised with provided.
Line 91: I think the question mark here should be a period.
Many thanks. We have revised.
Your discussion falls into 3 different categories, and it would be helpful to keep these separate (perhaps putting them in different sections of the paper): (1) Legal issues regarding the legal authority or “competence” of a regional legislative body to make this rule; (2) Administrative and medical concerns about how this rule could be put into practice; and (3) Ethical arguments about whether MAID (as described in this rule) is ethically permissible. At present, these 3 kinds of considerations are mingled together, and this reduces the clarity and effectiveness of your arguments.
Many thanks. We have extensively revised the text, also according to the other reviewers suggestions, to discuss with better clarity those aspects.
Lines 129-131: Although it’s not entirely clear here, you seem to be arguing that it would be very, very difficult for any patient to meet the requirement of experiencing suffering that is refractory to medical treatment. That is because one of the treatments available is deep palliative sedation – this would suspend all of a patient’s experiences, including any form of suffering. A standard view of continuous deep palliative sedation is that it should be used only when pain is intolerable and refractory to less extreme interventions and death is expected within a short time. (See: de Graeff A, Dean M (2007) Palliative sedation therapy in the last weeks of life: a literature review and recommendations for standards. J Palliative Med 10(1): 67-85.) Would a feeding tube be used in such cases?
Many thanks. The text was probably unclear. We have extensively revised the text, better distinguishing deep palliative sedation and MAID requirements and conditions. We have also discussed over life-supporting treatments.
Lines 132-134: Please clarify whether the difficulties you describe here are for the deep palliative sedation criteria, the MAID criteria, or both.
Many thanks. We have revised, as referring to both procedures. See section 2.3 Alternative options to MAID and deep sedative palliation (DPS).
There is a great deal of complexity in these issues that is being skimmed over. In assessing whether a treatment is ineffective, for instance, we need to clarify what that treatment is intended to accomplish, how that goal of treatment is related to the patient’s overall goals for treatment, what tolerance the patient (and physician) have for bad outcomes, which low-probability outcomes are worth pursuing, and so forth. The concept of proportionality is also complex and is often found in bioethics scholarship rooted in Roman Catholic theology. If these issues are important to your argument, present some explanation of each.
Many thanks. We have revised in the section 2.3 Alternative options to MAID and deep sedative palliation (DPS).
Lines 136-139: It’s not clear what you mean here. Your description of this law in section 3 did not include a requirement that all medical alternatives to MAID had been ruled out. Please clarify what the law actually requires and revise section 3 to reflect that.
Many thanks. We have integrated the text on new law procedure.
What is the commission’s role here? Does it make medical determinations about which treatments are viable alternatives to MAID? Does it assess whether enough alternatives have been tried and assess whether they have truly failed?
By the way, how would an alternative be ruled out? Must the patient actually try every treatment for which s/he is a medical candidate? What if a patient does not consent to one of these alternatives? Does that mean that they cannot meet the requirements for MAID? What if an alternative treatment is too expensive for the patient, or is not available within a reasonable distance? What about the option of voluntary cessation of eating and drinking? (See: Quill TE, Ganzini L, Truog RD, et al. (2018) Voluntarily Stopping Eating and Drinking Among Patients With Serious Advanced Illness—Clinical, Ethical, and Legal Aspects. JAMA Intern Med. 178(1):123-127.)
Lines 140-142: I thought deep palliative sedation was being presented as a difficulty in meeting the requirement that the patient be in intolerable, intractable pain (with the result that no one would ever meet the criteria for MAID). But here you’re addressing the suggestion that deep palliative sedation is an alternative to MAID – in other words, that it offers another means of dying in order to avoid further suffering. Has anyone taken that position? If so, it would be helpful to cite it. If not, why bother to argue against it?
Many thanks. The law is unclear on how ascertain informative disclosure and alternative options of MAID to patients. Moreover, the law seems to consider DPS an alternative to MAID while it’s a particular option in specific pathological conditions.
We have extensively revised the text, also according to your previous suggestions. See section 2.3 Alternative options to MAID and deep sedative palliation (DPS).
Lines 143-163: This paragraph is very unclear. It begins with a comparison of drugs used in MAID and continuous deep palliative sedation. How is this related to a discussion of problems with MAID? The paragraph then briefly addresses legal differences between active and passive euthanasia, and then turns to the complex issue of informed consent, decision-making competence, and advance directives. Please separate these issues, show how they are relevant to the topic of this paper, and address them in sufficient detail.
Many thanks. We have revised the paragraph. We have then discussed from a medico-legal point of view the difference between DPS (active medical act) and MAID (passive nursing act) to highlight the limited perimeter of practical implementation of the law.
Lines 157-163: The issue of informed consent is very important and very complex; it requires a much more detailed discussion.
We agree. We have briefly discussed the role on information and consenting regarding MAID and the role of living wills.
What are the legal standards for assessing decision-making capacity and what are the clinical tests used to make this determination? Would someone experiencing intolerable suffering still be competent to make this sort of life-and-death decision? Why or why not? Are there concerns in MAID cases about whether the patient’s decision is unduly influenced by factors such as the cost of treatment, the availability of treatment and supportive care, the attitudes of family members, concern about “being a burden” to family, etc.? What does the law say about whether a patient can use a living will or other advance medical directive to choose MAID?
Many thanks. We have extensively revised the text trying to briefly answer all these points, linked with the new regulation.
Lines 164-171: The waiting periods commonly found in MAID laws generally serve 2 purposes: (a) providing time for medical evaluation and consultation, competence assessment, exploring options, etc.; and (b) allowing an opportunity for the patient to reflect on this choice and perhaps decide against it. It is worth pointing out that we are dealing here with patients whose suffering is so great that they would rather die than continue to endure it. The longer we make them wait, the more avoidable suffering they experience. One of the ethical challenges, then, is to find the right balance between providing enough time for reflection of the sort associated with purpose (b) and minimizing suffering by keeping that time as short as possible. In the American state of Oregon, data shows that a number of people who start the process of seeking medical assistance in dying die from their illnesses before that process is complete. (See: https://www.oregon.gov/oha/ph/providerpartnerresources/evaluationresearch/deathwithdignityact/Documents/year26.pdf)
Many thanks, we agree on the issues related to the waiting time. We have discussed further, mentioning the need for balancing interests.
Other ethical issues associated with MAID have not been addressed: The implied devaluing of the lives of people with disabilities or chronic illnesses, the vulnerability for abuse or misuse, etc.
We understand also the relevance of those issues, but we have tried to focus on medico-legal, health management and health law perspective (those the authors), than bioethical ones.
The conclusion (section 5) mentions some issues that have received very little attention in this paper: The issue of access and medical migration, lessons to be learned from laws in other countries, and neurodegenerative or psychiatric diseases. Please expand on these topics in the main body of the paper.
Many thanks. We have briefly discussed in the introduction and last paragraph 2.5 Psychiatric conditions and MAID.
Comments on the Quality of English Language
The writing is sometimes awkward and needs clarification in some areas (as noted in my comments).
Many thanks. We have extensively revised the text.
Sincerely, livio tronconi
Round 2
Reviewer 2 Report
Comments and Suggestions for Authors
The revisions have greatly improved the clarity and quality of the discussion.
Comments on the Quality of English LanguageThere are some fairly minor issues with clarity, grammar, and usage. I recommend having someone proofread this manuscript. Otherwise, this is now ready for publication.